# iTRAQ-Based Quantitative Proteomic Analysis of Digestive Juice across the First 48 Hours of the Fifth Instar in Silkworm Larvae

**DOI:** 10.3390/ijms20246113

**Published:** 2019-12-04

**Authors:** Pingzhen Xu, Meirong Zhang, Ping Qian, Jiawei Li, Xueyang Wang, Yangchun Wu

**Affiliations:** 1School of Biotechnology, Jiangsu University of Science and Technology, Sibaidu Rd, Zhenjiang 212018, China; xpz198249@just.edu.cn (P.X.); zhangmr@just.edu.cn (M.Z.); qianping@just.edu.cn (P.Q.); 162212001131@just.edu.cn (J.L.); xueyangwang@just.edu.cn (X.W.); 2Sericulture Research Institute, Chinese Academy of Agricultural Sciences, Sibaidu Rd, Zhenjiang 212018, China

**Keywords:** *Bombyx mori*, iTRAQ, proteomics, digestive juice

## Abstract

The silkworm is an oligophagous insect for which mulberry leaves are the sole diet. The nutrients needed for vital activities of the egg, pupal, and adult stages, and the proteins formed in the cocoon, are all derived from the larval stages. The silkworm feeds and grows quickly during the larval stages. In particular, the amount of leaf ingested and digested quickly increases from the ecdysis to the gluttonous stage in the fifth instar period. In this study, we used the iTRAQ proteomic technique to identify and analyze silkworm larval digestive juice proteins during this period. A total of 227 proteins were successfully identified. These were primarily serine protease activity, esterase activity, binding, and serine protease inhibitors, which were mainly involved in the digestion and overcoming the detrimental effects of mulberry leaves. Moreover, 30 genes of the identified proteins were expressed specifically in the midgut. Temporal proteomic analysis of digestive juice revealed developmental dynamic features related to molecular mechanisms of the principal functions of digesting, resisting pathogens, and overruling the inhibitory effects of mulberry leaves protease inhibitors (PIs) with a dynamic strategy, although overruling the inhibitory effects has not yet been confirmed by previous study. These findings will help address the potential functions of digestive juice in silkworm larvae.

## 1. Introduction

The silkworm, *Bombyx mori*, is a complete metamorphosis insect. It is not only a domestic insect of economic significance, but also a model insect of Lepidoptera, which is widely used in basic and applied research [1,2]. The silkworm is an oligophagous insect and mulberry leaves are its sole diet [3]. The silkworm digestive system is a simple tube that runs longitudinally along the central body cavity from the mouth to the anus in larval stages. The digestive tube can be divided into the foregut, midgut, and hindgut according to its function and structure [4]. The main functions of the foregut are physical digestion, consisting of feeding, grinding, swallowing, and temporary storage of mulberry leaves. The hindgut is present in the midgut transition that is generally thought to reabsorb certain salts and amino acids from food residue to maintain water balance and/or osmotic pressure in the silkworm [5,6,7]. The midgut is the most important section and comprises approximately 78 percent of the total length of the digestive tube. The midgut originates from the endoderm that consists of muscle, basilar membrane, epithelium layer, and peritrophic envelope [8]. The main functions of the midgut are chemical digestion focused on digesting and absorbing with the action of enzymes in digestive juice.

Digestive juice is secreted by cylindrical and cup cells in the midgut. Cylindrical cells are the main components of epithelial cells and cup cells are mainly distributed in the interior of epithelial cells [9]. The silkworm digestive juice has an alkaline pH between 9.2 and 9.8, and the highly alkaline pH between 10.5 and 11.0 in the central midgut region. The gut pH is one of the most important regulators of digestive enzyme activity in insects [10]. The macromolecular nutrients must be degraded into small molecules by digestive enzymes before they can be absorbed by epithelial cells. There are many kinds of digestive enzymes with high activities in the midgut of silkworm. According to the location of these digestive enzymes, they can be divided into two categories. The first is exoenzymes that are synthesized by epithelial cells and involved in the degradation of macromolecular nutrients, such as trypsin, lipase, amylase, and nuclease. These enzymes can work efficiently under alkaline conditions. The other type is endoenzymes, which are mainly located in the microvilli of cylindrical cells. The most suitable pH value is almost neutral or faintly acidic, such as peptidase and oligosaccharide enzymes. The silkworm is oligophagous and consumes a significant quantity of fresh mulberry leaves, which have a high growth rate. Comparative proteomics analyses between mulberry leaves and silkworm feces indicate that large amounts of proteins from mulberry leaves are absorbed and digested by silkworm larvae [11]. The silkworm larval midgut has a complex proteolytic environment with different properties. Examining the enzyme activities in digestive juice, the amylase has a positive correlation with survivability, and the alkaline phosphatase and invertase has positive roles in the expression of yield attributes [12,13,14]. The esterases, specific β-esterase bands (Est-1, 2 and 3), have been documented as being present in digestive juice [14]. Catalyzing food protein hydrolysis is a crucial step in digestion. Serine proteases (SPs) that catalyze the hydrolysis of peptide bonds in proteins are the most abundant proteinase in digestion [15,16]. It has been confirmed that the digestive enzymes are all responsible for protein digestion, including trypsins, chymotrypsins, elastases, cathepsin B-like proteases, aminopeptidases, and carboxypeptidases [16].

Bacteria and viruses, excluding fungi, can infect *B. mori* larvae via the oral pathway. As larvae ingest the occlusion-derived virus (ODV) of nucleopolyhedrovirus (NPV), the alkaline digestive juice dissolves it and releases the enveloped virions, which then begin to infect the midgut columnar epithelial cells [16]. The roles of many proteins in digestive juice to resist pathogens have been investigated in silkworms. The novel red fluorescent protein (RFP) purified from silkworm digestive juice possesses antiviral, antifungal, and antibacterial properties [17]. Multiple forms of RFPs (A, B and C) exhibit unique specificity in neutralizing the different viruses, namely NPV, cypovirus (CPV), infectious flacherie virus (IFV), and densovirus (DNV) to different degrees [18]. RFP was detected only in the digestive juice and was not detected in the hemolymph [17]. Light is essential in the synthesis of RFPs in *B. mori*-fed mulberry leaves. RFPs have been reported to inactivate NPV and the molecular weights of each are significantly different [18,19,20]. Different varieties of silkworms have varying numbers of RFPs that are related to the susceptibilities of the silkworms to viral disease [21]. Serine proteases (SPs) and serine protease homologs (SPHs) showing strong antiviral activity in the digestive juice of silkworm have been previously reported [16,22,23].

The amount of leaf ingested and digested increases daily from the first to the third days of the fifth instar silkworm larva (before the gluttonous stage). In this period, the risk of silkworm infection is potentially raised. Therefore, the silkworm may need higher digestive enzyme activity and more active immune activity. However, the previous studies investigating the digestive juice proteins have not been conducted at the whole proteomics level. Moreover, the components of digestive juice proteins have not been analyzed in different developmental periods. In the present study, we used the iTRAQ quantitative proteomic technique to identify and analyze silkworm digestive juice proteins in different developmental periods. This work first focused on the complex composition of silkworm digestive juice, in order to understand the complex biological processes in nutrient digestion that are very important before the silkworm larvae progresses to the gluttonous stage.

## 2. Results

### 2.1. Protein Profiling

After processing all MS/MS spectra in Mascot software, 1126 unique tryptic peptides were mapped to 227 proteins from silkworm digestive juice. The unique tryptic peptides defining each protein and 227 successfully identified proteins are shown in Appendix A, including the information of each protein relating to the locus name of a gene in the silkworm genome, each protein accession number from the NCBInr database, the probe number from the silkworm microarray database, the predicted signal peptide, the functional description, the identified percentage of total amino acid sequence, and the number of unique peptides per protein. A total of 182 of the 227 identified proteins had signal peptides. The 227 proteins were identified in digestive juice from ecdysis to gluttonous stages in the fifth instar period. The amount of leaf ingested and digested by silkworm larvae most quickly increases in this period, such that it may need higher digestive enzyme activity and more immune activity in digestive juice. Remarkably, among the 227 identified proteins, there were 20 serine proteases, 21 serine protease inhibitors, 9 carboxypeptidases, 9 aminopeptidases, 7 lipases, 6 lipoproteins, 4 collagenases, 6 immune-related proteins, 12 enzymes involved in nucleic acid metabolism, 10 enzymes involved in sugar metabolism, and 22 uncharacterized proteins (Appendix A). In addition, the isoelectric points of 70 proteins were between 7.01 and 11.94 (Appendix A).

### 2.2. Identification of Differentially Expressed Proteins

The threshold for up- and downregulation of a fold change of ≥ 2.0 or ≤ 0.5 was found to be differentially expressed in digestive juice of the three developmental stages in comparison with each other. In DJ24h (feeding 24 h, day two of fifth instar) compared with DJ0h (feeding 0 h, day one of fifth instar), 43 proteins were upregulated and 28 proteins were downregulated (Figure 1A and Table 1, Appendix A). In DJ48h (feeding 48 h, day three of fifth instar) compared with DJ0h, 63 proteins were upregulated and 50 proteins were downregulated (Figure 1A and Table 1, Appendix A). Thus, it can be seen that the number of differentially expressed proteins (DEPs) increases with the feeding time. A total of 27 proteins were upregulated and 40 proteins were downregulated in DJ48h compared with DJ24h (Figure 1A and Table 1, Appendix A). In total, there were 11 DEPs across the comparison of the three groups (Figure 1B). The functions of upregulated expression proteins were mainly focused on protein and carbohydrate metabolism, and serine protease activity in DJ24h compared with DJ0h; serine protease activity and protein and carbohydrate metabolism in DJ48h compared with DJ0h; and protein and carbohydrate metabolism in DJ48h compared with DJ24h (Table 1, Appendix A). Moreover, the functions of downregulated expression proteins were primarily located in the binding proteins of ions, fatty acids, and chitin in DJ24h compared with DJ0h; serine protease inhibitor and binding proteins of ions, fatty acids, and chitin in DJ48h compared with DJ0h; and serine protease inhibitor and serine protease activity in DJ48h compared with DJ24h (Table 1, Appendix A).

A total of nine carboxypeptidases were successfully identified in silkworm digestive juice; among them, BGIBMGA004830, BGIBMGA009486 and BGIBMGA004797 were upregulated in DJ48/24h (Table 1, Appendix A). Meanwhile, nine aminopeptidases were also successfully identified; BGIBMGA008061 and BGIBMGA001640 were upregulated in DJ24/0h, BGIBMGA008017 and BGIBMGA008059 were upregulated in DJ48/0h, and BGIBMGA009138 was upregulated in DJ48/24h (Table 1, Appendix A). Moreover, five serine proteases associated with immune response during *Escherichia coli* and *Bacillus bombyseptieus* infection were identified (Table 1, Appendix A); BGIBMGA003604 and BGIBMGA010276 were upregulated in DJ24/0h, BGIBMGA003568, BGIBMGA003566 and BGIBMGA008514 were upregulated in DJ48/0h. The abundance of innate immune pattern-recognition receptor beta-1,3-glucan recognition protein 4 (BGIBMGA000353) was increased daily from day one to day three of the fifth instar larvae in digestive juice (Table 1). These results suggest that the digestion and immunity of *B. mori* larvae are gradually enhanced.

### 2.3. Gene Ontology and Kyoto Encyclopedia of Genes and Genomes Pathway Enrichment Analysis of DEPs

After obtaining the differentially expressed proteins (DEPs), they were annotated by Gene Ontology (GO) analysis to be involved in biological processes, molecular functions, and cellular components (Figure 2). The DEPs identified in DJ24h compared with DJ0h were significantly enriched for proteins related to biological processes that were basically focused on metabolic processes and cellular processes (Figure 2A, Appendix A). With regard to the molecular function, proteins were involved in catalytic activity and binding (Figure 2A, Appendix A). Within the cellular component, the DEPs were primarily located in the extracellular region (Figure 2A, Appendix A). In DJ48h compared with DJ24h, regarding biological processes, the DEPs were mainly focused on metabolic processes, single-organism processes, and cellular processes; for the molecular function, proteins were involved in catalytic activity and binding; and within the cellular component, proteins were primarily located in the extracellular region (Figure 2B, Appendix A). There were some differences in the GO functional annotation of the differentially expressed proteins from the two comparison groups (Figure 2). Kyoto Encyclopedia of Genes and Genomes (KEGG) pathway enrichment analysis of the identified DEPs showed that the enriched proteins were mainly involved in pathways, including galactose metabolism, sphingolipid metabolism, folate biosynthesis, ubiquitin mediated proteolysis, and purine metabolism in DJ24h compared with DJ0h, and their functions were related to metabolisms, development, regulation of innate immune signaling, and antiviral properties (Table 2, Appendix A). Meanwhile, in DJ48h compared with DJ24h, the enriched proteins were mainly involved in pathways, including folate biosynthesis, lysosome, galactose metabolism, fructose and mannose metabolism, pentose and glucuronate interconversions, purine metabolism, and peroxisome (Table 3, Appendix A).

### 2.4. Protein–Protein Interaction Network Analysis of DEPs

For further analysis, DEPs were submitted to the STRING database to generate protein-protein interaction networks. The main results of the enrichment network indicated that Ubi3 comprised the central node of protein-protein interactions (Figure 3). Swiss-Prot accession number H9IVY4 was consistent with BGIBMGA001415, which encoded polyubiquitin-A protein (Ubi3) with the full length of 913 aa. The Ubi3 contained twelve ubiquitin (UBQ) domains predicted by SMART online. Its GO function was protein binding (GO:0005515). Ubiquitin is mainly fused with ribosomal proteins [44,45]. Multiple ribosomal proteins interacting with Ubi3 were found in the STRING database (Appendix A). However, there were H9JYG4 (BGIBMGA009648, histone H), H9JK60 (BGIBMGA009911, ubiquitin-activating enzyme E1), O97158 (BGIBMGA011424, transferrin), and Q86D78 (BGIBMGA003512, beta-glucosidase) interacting with Ubi3 among all the DEPs in digestive juice, respectively (Figure 3). These results suggest that Ubi3 may be involved in the digestion process of the developmental periods.

### 2.5. Spatial Expression Profile of Genes of the Identified Proteins

Day three of the fifth instar of the silkworm is the boundary for whole larval development stage [2]. The fifth instar of silkworm larva feeds and grows quickly before this time, but afterward, the silkworm hemolymph proteins gradually change and silkworms to synthesize massively silk proteins in the silk gland [2,46]. In order to investigate the spatial expression profile of genes of the identified proteins in digestive juice, the microarray data were downloaded from SilkDB, including the gene expression level across 10 *B. mori* larval tissues of day three of the fifth instar [2]. Thus, the study of this point in time will be helpful to elucidate the synthesis and secretory mechanism of digestive juice proteins. Genes of the identified digestive juice proteins had expression signals in at least one tissue (Figure 4, Appendix A, Appendix A). A total of 21 genes had expression signals in all 10 tissues (Appendix A, Appendix A). Most of the genes had high expression features in the midgut. Thirty genes expressed specifically in the midgut (Figure 4A, Appendix A). 57 genes expressed significant high in the midgut (Figure 4B, Appendix A). The midgut-specific genes encoded enzymes, hydrolase, and binding proteins that were involved in the digestion of mulberry leaves by GO functional annotation. These diverse expression characteristics suggest that digestive juice proteins are mainly secreted by digestive tube and the majority of the enzymatic digestion largely takes place in the midgut.

### 2.6. Expression Analysis of Genes Corresponding to DEPs

In the previous section, the genes of the identified digestive juice proteins were mainly expressed in the midgut. Thus, we used midgut tissue to verify whether the changes in protein level were consistent with the transcriptional level and also to examine the proteomics data. The three time-points of midgut and digestive juice were identical. RT-qPCR was used to investigate expression levels of genes of the DEPs at the three time-points. The information of the selected differentially expressed genes and *B. mori* ribosomal protein gene *BmRPL3* primers are presented in Appendix A. The eight genes for encoding four up- and downregulated proteins were selected, respectively. The changes of the eight genes’ transcription were generally consistent with their encoding proteins in the iTRAQ data (Figure 5).

## 3. Discussion

Silkworm larvae grow rapidly and the weight of the terminal fifth instar larvae is approximately 10,000 times that of a newly hatched larvae. Only the larvae feed during the whole lifecycle. The nutrients needed for the vital activities of egg, pupal, and adult stages, and the proteins formed in the cocoon, are derived from the larval stage. The core value of the larval stage is feeding, absorption, and accumulation of nutrients. In particular, the amount of leaf ingested and digested quickly increases from immediately after ecdysis to the gluttonous stage in the fifth instar period. At the same time, the risk of oral infection is potentially raised. Therefore, the characteristics of the ingredients and variation of silkworm digestive juice proteins are of interest.

In this study, we used iTRAQ to analyze proteomes in silkworm digestive juice and 227 proteins were successfully identified. The classifications of the whole identified proteins were mainly serine protease activity, esterase activity, binding, and serine protease inhibitor, which were mainly involved in the digestion and overruling the detrimental effects of mulberry leaves. The spatial expression profiles of the genes of the identified digestive juice proteins showed that most of the genes had high expression features in the midgut and 30 genes expressed in the midgut displayed tissue-specificity. The silkworm feeds and grows quickly from day one to day three of the fifth instar [46]. The abundance of identified proteins in day one, two, and three of fifth silkworm instar digestive juice was compared. In DJ24h compared with DJ0h, 43 proteins were upregulated and 28 proteins were downregulated; in DJ48h compared with DJ0h, 63 proteins were upregulated and 50 proteins were downregulated; and 27 proteins were upregulated and 40 proteins were downregulated in DJ48h compared with DJ24h. These results indicate that the digestive juice comprises a dynamically changing protein mixture in the developmental periods. Day three of the fifth instar is the boundary for whole larval development; after this time, silkworm hemolymph proteins around 30 kDa and 80 kDa gradually increase, and silkworms to synthesize mass silk proteins in the silk gland [2,47,48]. Therefore, silkworms need to strongly increase their digestive efficiency.

The silkworm needs to acquire essential nutrients from mulberry leaves by digesting dietary proteins, carbohydrates and lipids. Proteins are digested into amino acids and oligopeptides by digestive enzymes [10]. Trypsins, chymotrypsins, elastases, cathepsin-B like proteases, aminopeptidases, and carboxypeptidases are all responsible for protein digestion in lepidopteran larvae [15,16]. The exopeptidases are further divided into carboxypeptidases (cleaving at the carboxylic terminus) and aminopeptidases (cleaving at the amino terminus). A total of 48 genes encoding carboxypeptidases have been identified in silkworm and 11 midgut-specific carboxypeptidases were significantly downregulated after starvation and restored after re-feeding [49]. A total of 9 carboxypeptidases were successfully identified in silkworm digestive juice; among them, BGIBMGA004830, BGIBMGA009486, and BGIBMGA004797 were upregulated in DJ48/24h. Meanwhile, nine aminopeptidases were also successfully identified; BGIBMGA008061 and BGIBMGA001640 were upregulated in DJ24/0h, BGIBMGA008017 and BGIBMGA008059 were upregulated in DJ48/0h, and BGIBMGA009138 was upregulated in DJ48/24h. Serine proteases include trypsins, chymotrypsins, and elastases, which are known to dominate the larval gut environment and contribute to the majority (as much as 95%) of the total proteolytic activity in the gut of larval Lepidoptera [10,15]. The sources of mulberry leaf sugars are sucrose, starch, and reducing sugars, such as glucose and fructose [50]. Only monosaccharides of the end products of carbohydrate digestion are absorbed and utilized. The sugar absorbed and metabolized is essential for silkworm larval development and cocoon production [51]. The initial digestion of carbohydrates is mediated by amylases and only alpha-amylases have the activity to catalyze starch and glycogen in insects [10,52]. BGIBMGA001876 (silkworm alpha-amylase) was upregulated in DJ48/0h and DJ48/24h. Smaller polysaccharides and disaccharides produced by catalysis of α-amylase are further degraded to monosaccharides during subsequent steps of carbohydrate digestion performed by distinct enzymes [10,53]. After the initial breakdown of large oligosaccharides by amylases, α-glucosidase and β-fructofuranosidase mediate the subsequent degradation of oligo- and disaccharides into monosaccharides [10,51]. Sugar mimicking alkaloids, such as 1,4-dideoxy-1,4-imino-D-arabinitol (D-AB1) and 1-deoxynojirimycin (DNJ) derived from mulberry latex, are strong inhibitors of α-glucosidase but do not exhibit inhibitory activity against β-fructofuranosidase (β-FFase) [54,55]. BGIBMGA005696 (BmSuc1) was upregulated in DJ48/0h and DJ48/24h that is a novel animal β-FFase cloned and identified in *B. mori* [51,56]. BmSuc1 acts as an essential sucrase by directly modulating the degree of sucrose hydrolysis and silencing *BmSuc1*, significantly reducing glucose in the midgut and leading to smaller body size, lighter weight, and developmental delays of silkworm larvae [51].

Lipases are involved in lipid digestion and degrade dietary lipids to generate typical end products, such as free fatty acids, glycerols, partial acylglycerols, and phospholipid derivatives, in a process called lipolysis [10]. Lipases are active inside the midgut of silkworm larvae [57]. In this study, seven lipase related proteins were successfully identified. BGIBMGA001507 (Pancreatic triacylglycerol lipase, PNLIP) was upregulated in DJ48/0h and DJ48/24h. All insects use lipids for energy storage in the fat body [10,58]. Lipids contribute to 40% of the dry weight of an insect egg and are the most important supply of energy for the developing embryo [59]. Lipids are also incorporated in the silk gland synthetic activity and are synthesized to secrete into the anterior silk gland to promote silk spinning in silkworm larvae [60].

In addition, other proteins had been successfully identified, such as BGIBMGA001415 (Polyubiquitin-A protein, Ubi3), BGIBMGA001173 (BmdsRNase), BGIBMGA000353 (beta-1,3-glucan recognition protein 4, BmβGRP4), BGIBMGA012864 (Peptidoglycan recognition protein S3, BmPGRP-S3), and BGIBMGA002288 (Lipopolysaccharide binding protein, BmLBP). Ubi3 is mainly fused with ribosomal proteins [44,45]. The interaction of Ubi3 and BGIBMGA003512 (beta-glucosidase) protects beta-glucosidase from 1-deoxynojirimycin (DNJ) strong inhibitor in digestive juice, which may indicate they are all involved in the digestion process [55]. BmdsRNase is able to digest ssRNA, ssDNA, and dsDNA, and has highest activity towards dsRNA, which is involved in the innate immune response against viral invasion [61,62]. The BGIBMGA000353 (BmβGRP4), BGIBMGA012864 (BmPGRP-S3), and BGIBMGA002288 (BmLBP) are all innate immunity-related proteins in the silkworm [63]. BmLBP induces nodule formation and participates in clearance of bacteria by binding to gram-negative bacteria via lipopolysaccharide [64]. BGIBMGA012864 (BmPGRP-S3) plays a role in the immune response of the silkworm to *B. mori* cytoplasmic polyhedrosis virus (BmCPV) infection [65]. The β-1,3-glucan recognition protein (βGRP) family contains two functionally different proteins, one in combination with β-1,3-glucan, and the other dubbed gram-negative binding protein (GNBP) in combination with gram-negative bacteria or gram-positive bacteria [63]. βGRPs (GNBPs) trigger an innate immune response by activating the prophenoloxidase cascade and the Toll receptor pathway in hemocytes [66]. Four *BmβGRPs* have been identified in *B. mori* and BmβGRP3 specifically recognizes a triple-helical structure of β-1,3-glucan [63,67]. BmβGRP1-3 lack β-1,3-glucanase activity because the active catalytic residues for β-1,3-glucanase of BmβGRP1-3 are replaced with other amino acids; BmβGRP4 has the active catalytic residues [63,68]. The spatial expression profile of *BmβGRP4* was highest in the midgut, much lower but detectable in the testis and ovary, and undetectable in the remaining larval body tissues, including the mainly innate immune locations, such as hemocytes and the fat body, during day three of the fifth instar larvae. Moreover, the abundance of BGIBMGA000353 (BmβGRP4) was increased daily from day one to day three of the fifth instar larvae in digestive juice. *βGRP* has been identified in other lepidopteran insects. βGRP is essential to gut immunity for resistance against fungal pathogen and opportunistic pathogenic gut bacteria in *Locusta migratoria manilensis* [69]. βGRP-1 is purified and characterized from the midgut lumen of *Helicoverpa armigera*; the mRNA of *βGRP-1* is predominantly expressed in the midgut and is induced by feeding gram-negative or gram-positive bacteria, and the protein is secreted into the lumen that persist there in a stable manner [70]. βGRP-1 does indeed possess β-1,3-glucanase activity and functions catalytically in digestion and/or pathogen defense in *H. armigera* [70]. The βGRPs persisting in larval digestive juice are related to but distinct from the previously described βGRP/GNBP proteins primarily found in lepidopteran hemolymph, which do not possess β-1,3-glucanase activity. The tissue-specific proteins often exhibit a strong relevance to the physiological functions of the corresponding tissues. The functional characteristics of this new midgut-specific βGRP protein suggest that lepidopteran larvae secrete an active β-1,3-glucanase into digestive juice in order to digest β-1,3-glucans released by commensal or invading bacteria and protect midgut cells from microbial invasion [70]. Bacteria and viruses, excluding fungi, can infect *B. mori* larvae via oral pathway. The roles of many proteins in digestive juice to resist pathogens have been investigated in silkworms. The presence of antiviral proteins, such as multiple forms of red fluorescent protein (RFPs) [17], Bmlipase-1 [22,71], dsRNase [61,62], NADPH oxidoreductase (BmNOX) [72], serine proteases (SPs), and serine protease homologs (SPHs) [16], have been previously reported. Meanwhile, RFPs, SPs, and SPHs also have antibacterial functions.

## 4. Materials and Methods

### 4.1. Collection of Samples and Protein Preparation

Larvae of *B. mori* strain “radiation seven” were reared on mulberry leaves at a stable temperature of 25 °C. Digestive juice of 90 larvae per group was collected from first day (immediately after ecdysis, feeding 0 h, referred DJ0h), second day (feeding 24 h, referred DJ24h) and third day (feeding 48 h, referred DJ48h) of fifth instar as previously described [9]. The midgut corresponding to the same time-point of the three stages was also collected to verify gene expression analysis. Then, each sample of digestive juice was filtered through a 0.22-μm filter and quantitated by the Bradford assay kit [73]. A quantity of 20 μg of each protein sample was separated by SDS-polyacrylamide gel electrophoresis (SDS-PAGE).

### 4.2. Protein Digestion and iTRAQ Labeling

Protein digestion was carried out according to previous protocols as described [74,75]. Finally, the protein suspension was digested with 40 μL of trypsin buffer containing 3 μg trypsin (Promega, Madison, WI, USA) at 37 °C for 16–18 h. Filtrate was collected and peptides were quantitated at OD_280_ [74]. Subsequently, the digested peptides were labeled with iTRAQ reagents following the manufacturer’s instructions (Applied Biosystems, Framingham, MA, USA) using 114-tag, 115-tag, and 116-tag for feeding 0, 24 and 48 h samples (named Dj0h, Dj24h and Dj48h), respectively.

### 4.3. Strong Cation Exchange Fractionation and NanoLC-MS/MS Analysis

Strong cation exchange (SCX) chromatography of the mixed samples and nanoLC-MS/MS analysis were performed as previously described [9,75]. Separated samples were analyzed with a Q Exactive (Thermo Fisher Scientific, Waltham, MA, USA) that was coupled to an Easy nLC 1200 system (Thermo Fisher Scientific, Waltham, MA, USA) for 120 min to identify peptides. Detection mode was positive ion, mother ion scanning range of 300–1800 m/z. One-level mass spectrometry resolution was 70,000 at m/z 200. The automatic gain control (AGC) target was set to 3e6, and maximum inject time (IT) to 10 ms. The number of scan ranges was 1. Dynamic exclusion duration was 30.0 s. Mass-charge ratios of peptides and peptide fragments were collected by the following methods. Ten fragment maps were collected after each full scan. The resolution for HCD spectra was set to 17,500 at 200 m/z, and an isolation width of 2 m/z. Normalized collision energy was 30 eV. The under fill ratio was defined as 0.1%. The instrument was operated under the condition that the peptide recognition mode was allowed.

### 4.4. Protein Identification and Annotation

The protein identifications were performed using Proteome Discoverer 1.4 software (Thermo Fisher, Waltham, MA, USA) running Mascot 2.2 (Matrix Science, London, UK) by searching against the silkworm database (http://www.silkdb.org/silkdb/doc/download.html) and UniProt database of the silkworm, *Bombyx mori*. Database search parameters were peptide mass tolerance: 20 ppm; fragment mass tolerance: 0.1 Da; enzyme: trypsin; max missed cleavages: 2; protein and peptide FDR ≤ 0.01 [76]. A minimum of 1 unique peptide was used for the confident protein identification. For protein quantization, a protein must contain at least two unique peptides. The quantitative protein ratios were weighed and normalized by dividing via the average value of all peptides identified. The threshold for up/downregulation of fold change of ≥2.0 or ≤0.5 was used to consider statistically significant differences in abundance. After obtaining the differentially expressed proteins (DEPs), protein–protein interaction network analysis was defined with STRING software (http://string-db.org/) [77], and functional annotation was performed using the gene ontology (GO) assignments [78,79] and Kyoto Encyclopedia of Genes and Genomes (KEGG) pathway enrichments [80].

### 4.5. Tissue Expression Patterns Based on Microarray Data

In order to analyze tissue expression patterns of genes encoding the identified proteins, the microarray data were downloaded from the SilkMDB [2]. The microarray data of the DEP genes are provided in Appendix A. The microarray data are raw. The mean value of the microarray data of genes expressed specifically in the midgut is provided in Appendix A. GeneCluster 2.0 software was used to visualize the expression levels [81]. The expressed genes are defined as previously described [82].

### 4.6. RT-qPCR Analysis

The genes selected according to protein expression detected by iTRAQ were compared by reverse transcription-quantitative PCR (RT-qPCR) at the transcriptional level. Total RNA from the midgut samples of the three stages (named Mg0h, Mg24h and Mg48h) was used to synthesize the first strand cDNA using the PrimeScript Reverse Transcriptase kit (TaKaRa, Dalian, China) according to the instructions of the manufacturer. RT-qPCR was performed as we previously described [83]. The information of the selected differentially expressed genes and *B. mori* ribosomal protein gene *BmRPL3* primers are presented in Appendix A.

## 5. Conclusions

The silkworm, *Bombyx mori*, is a complete metamorphosis insect. The nutrients needed for the vital activities of the egg, pupal, and adult stages, and proteins formed in the cocoon, are all derived from the larval stages. The silkworm feeds and grows quickly during larval stages. In particular, the amount of leaf ingested and digested most quickly increases from ecdysis to gluttonous stages in the fifth instar period. In this study, we used the iTRAQ proteomic technique to identify and analyze silkworm larval digestive juice proteins in this period. A total of 227 proteins were successfully identified. Among the 227 proteins, the isoelectric points of 70 proteins were between 7.01 and 11.94; 30 genes of the identified proteins were of tissue-specific expression in the midgut. The number of differentially expressed proteins (DEPs) increased with the feeding time, as found via comparing them in chronological order. Temporal proteomic analysis of digestive juice revealed developmental dynamic features related to molecular mechanisms of the principal functions of digesting, resisting pathogens, and overruling the inhibitory effects of mulberry leaves protease inhibitors (PIs) with a dynamic strategy. To our knowledge, this study is the first report on the identification of possible proteins in silkworm larval digestive juice from the ecdysis to gluttonous stages in the fifth instar period. Our findings expand the current knowledge of the complex biological processes in nutrient digestion and provide a new perspective on the developmental dynamic features related to molecular mechanisms of digestive juice in silkworm fifth instar larvae.

## Figures and Tables

**Figure 1 ijms-20-06113-f001:**
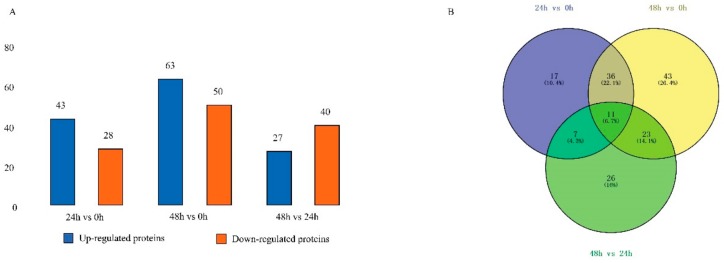
Identification and analysis of the differentially expressed proteins (DEPs) in digestive juice of the three developmental stages. (**A**) Up- and downregulated DEPs in digestive juice in the three stages. (**B**) DEP distribution Venn diagram.

**Figure 2 ijms-20-06113-f002:**
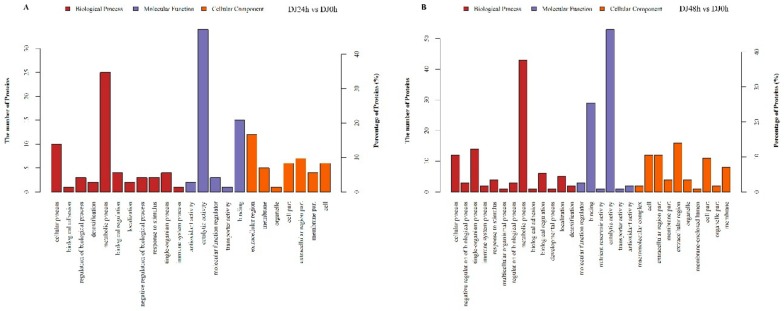
Gene ontology enrichment analysis for DEPs. Proteins are annotated by biological process, cellular component, and molecular function. (**A**) The group of DJ24h (feeding 24 h) compared with DJ0h (feeding 0 h). (**B**) The group of DJ48h (feeding 48 h) compared with DJ24h (feeding 24 h).

**Figure 3 ijms-20-06113-f003:**
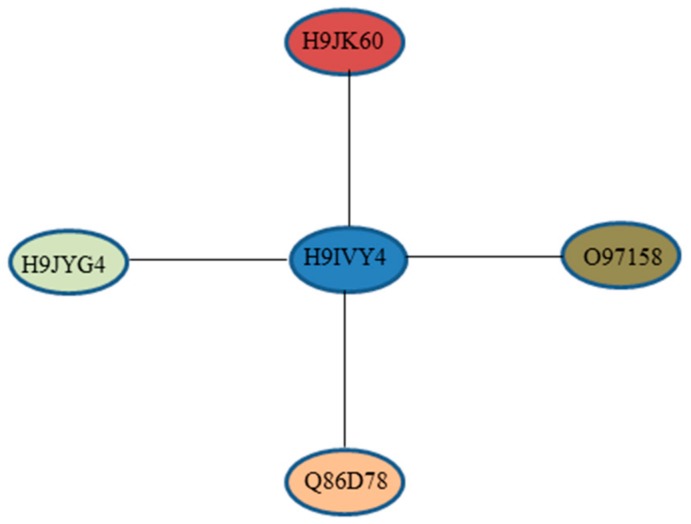
Protein–protein interaction networks of DEPs. Prediction and analysis of protein–protein interaction networks of all the DEPs was supported by STRING. H9IVY4: BGIBMGA001415, polyubiquitin-A, protein binding (GO:0005515). H9JYG4: BGIBMGA009648, histone H3, DNA binding (GO:0003677). H9JK60: BGIBMGA009911, ubiquitin-activating enzyme E1, protein binding (GO:0005515). O97158: BGIBMGA011424, transferrin, iron ion binding (GO:0005506). Q86D78: BGIBMGA003512, glucosidase, hydrolase activity (GO:0016787).

**Figure 4 ijms-20-06113-f004:**
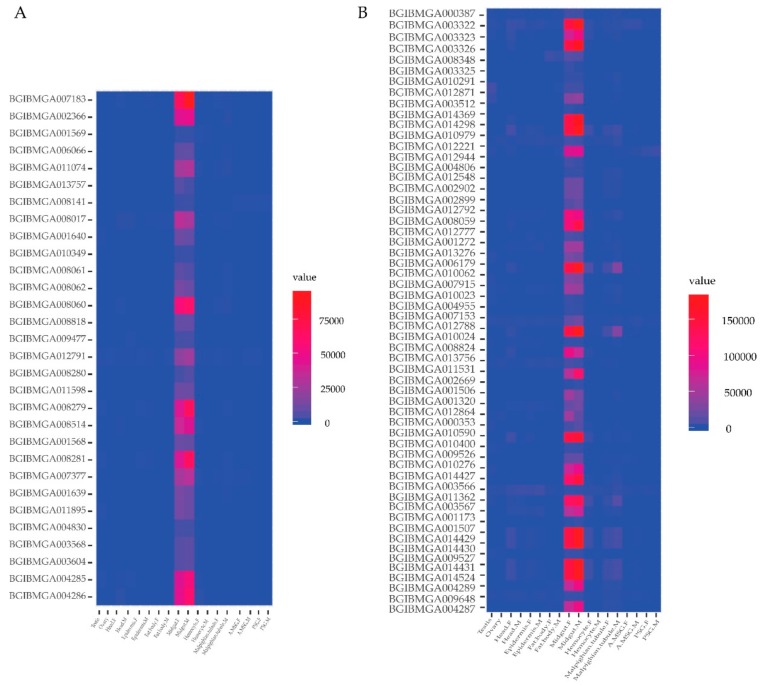
Tissue expression profile of genes of the identified digestive juice proteins in larvae on day three of the silkworm fifth instar. (**A**) The genes expressed specifically in the midgut. (**B**) The genes expressed significant high in the midgut. The columns represent ten different tissues with both sexes: testis, ovary, head, epidermis, fat body, midgut, hemocyte, Malpighian tubule, anterior/median silk gland (A/MSG), posterior silk gland (PSG), female (F), and male (M). Gene expression levels are represented by red (higher expression) and blue (lower expression) boxes. The mean value of the microarray data of each gene is calculated in each tissue.

**Figure 5 ijms-20-06113-f005:**
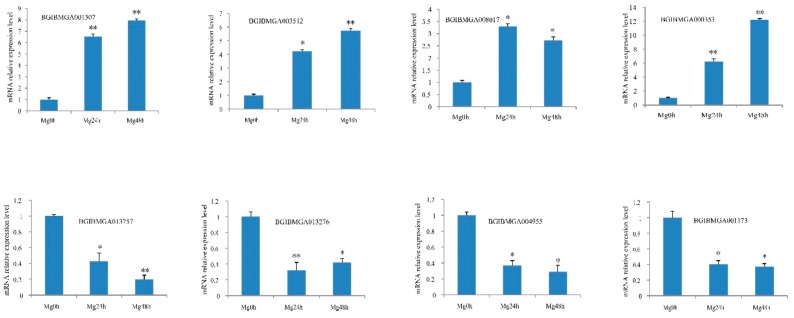
Analysis of changes in transcription level via RT-qPCR corresponding to the DEPs. For each gene, the mRNA level at Mg0h (midgut from day one of fifth instar larvae) is set as 1. The experiments were repeated three times. *B. mori* ribosomal protein gene *BmRPL3* is used as an internal control. The data are the means ± SD of three independent experiments. The significant differences are indicated by * (*p* < 0.05) or ** (*p* < 0.01).

**Table 1 ijms-20-06113-t001:** List of the proteins upregulated in silkworm digestive juice at 24 h and 48 h growth stages.

Time (h)	Number	Accession	SilkDB Accession	Fold Change	Function
DJ24/0 h	1	H9J8I4	BGIBMGA005826	2.11	Alcohol dehydrogenase activity
	2	H9IW77	BGIBMGA001508	2.19	Carboxylic ester hydrolase activity
	3	Q58I81	—	4.18	Ester bonds hydrolase activity
	4	H9JTK9	BGIBMGA012871	2.02	Ester bonds hydrolase activity
	5	H9ISX5	BGIBMGA000353	3.24	Innate immune
	6	Q86D78	BGIBMGA003512	2.72	Beta-glucosidase
	7	H9JCF2	BGIBMGA007153	2.73	Glycosyl hydrolase activity
	8	H9JL39	BGIBMGA010240	3.96	Chitinase activity
	9	H9JSN7	BGIBMGA012548	3.21	Sphingomyelin phosphodiesterase activity
	10	B1Q138	—	2.37	Carboxylesterase activity
	11	H9JTY2	BGIBMGA012994	3.01	CN_hydrolase activity
	12	B1Q137	—	3.08	Carboxylesterase activity
	13	H9IWD7	BGIBMGA001568	2.57	Maltase-glucoamylase activity
	14	B2ZZX0	BGIBMGA008818	4.87	Phosphatase activity
	15	H9JT78	BGIBMGA012740	2.82	Peroxidase activity
	16	H9JBZ7	BGIBMGA007042	2.92	Peroxidase activity
	17	P10831	—	3.33	Peptidase regulator
	18	Q03383	—	2.02	Antichymotrypsin
	19	C0J8G5	BGIBMGA003292	3.63	Serine protease inhibitor
	20	H9IXK0	BGIBMGA001983	10.24	Serine protease inhibitor
	21	H9JH30	BGIBMGA008827	13.22	Serine protease inhibitor
	22	I3VR74	BGIBMGA008061	2.53	AMP deaminase activity
	23	B5TZ28	BGIBMGA007915	4.92	Metallopeptidase activity
	24	H9J232	BGIBMGA003569	2.96	Serine protease activity
	25	H9J267	BGIBMGA003604	3.19	Serine protease activity
	26	H9JL75	BGIBMGA010276	3.15	Serine protease activity
	27	H9JJ25	BGIBMGA009526	4.40	Serine protease activity
	28	H9JJ26	BGIBMGA009527	3.72	Serine protease activity
	29	H9IWJ8	BGIBMGA001630	2.32	Cholinesterase activity
	30	H9JDE9	BGIBMGA007546	2.19	Carboxylesterase activity
	31	H9J067	BGIBMGA002902	2.84	Cholinesterase activity
	32	H9IWK8	BGIBMGA001640	2.89	Aminopeptidase activity
	33	H9JJ19	BGIBMGA009520	6.95	WD repeat domain phosphoinositide-interacting
	34	Q1HPY8	—	4.25	Guanine nucleotide binding
	35	H9J1D5	BGIBMGA003322	2.28	Juvenile hormone binding
	36	H9JK60	BGIBMGA009911	2.05	Ubiquitin-activating
	37	H9IVY4	BGIBMGA001415	2.08	Ubiquitin-mediated protein binding
	38	C1K001	BGIBMGA004287	5.34	N/A
	39	H9JN76	BGIBMGA010979	2.20	N/A
	40	H9J1D8	BGIBMGA003325	3.80	N/A
	41	H9JXD9	BGIBMGA014204	2.36	N/A
	42	H9JPZ3	BGIBMGA011598	2.39	N/A
	43	H9J1D9	BGIBMGA003326	2.33	N/A
DJ48/0 h	1	H9IW76	BGIBMGA001507	5.27	Lipase
	2	C7EPE2	BGIBMGA000158	5.42	Glucose-methanol-choline (GMC) oxidoreductase activity
	3	H9JFH1	BGIBMGA008268	3.06	Aldehyde oxidase activity
	4	H9J8I4	BGIBMGA005826	3.90	Alcohol dehydrogenase activity
	5	H9JTY7	BGIBMGA012999	2.94	Glucose dehydrogenase activity
	6	H9ISL2	BGIBMGA000239	3.87	Peroxidase activity
	7	H9ISL1	BGIBMGA000238	7.03	Peroxidase activity
	8	H9JT78	BGIBMGA012740	3.46	Peroxidase activity
	9	H9IX93	BGIBMGA001876	2.78	Alpha-amylase activity
	10	Q86D78	BGIBMGA003512	3.44	Beta-glucosidase
	11	B2DD57	BGIBMGA005696	2.81	Glycosyl hydrolase activity
	12	H9JCF2	BGIBMGA007153	3.18	Glycosyl hydrolase activity
	13	A0A077JI83	BGIBMGA006066	3.90	O-Glycosyl hydrolase activity
	14	H9ISX5	BGIBMGA000353	8.99	Innate immune
	15	H9JPS6	BGIBMGA011531	2.85	Phospholipase C activity
	16	H9IVS0	BGIBMGA001351	3.11	Oxidoreductase activity
	17	B2ZZX0	BGIBMGA008818	2.74	Phosphatase activity
	18	Q9NGS0	—	3.05	N/A
	19	P81902	—	3.16	Trypsin inhibitor
	20	Q03383	—	3.14	Antichymotrypsin
	21	C0J8G5	BGIBMGA003292	7.33	Serine protease inhibitor
	22	C0J8H1	—	12.35	N/A
	23	C4B489	BGIBMGA004445	2.19	Serine protease activity
	24	H9JFI3	BGIBMGA008280	3.17	Serine protease activity
	25	H9J229	BGIBMGA003566	2.04	Serine protease activity
	26	H9JKH3	BGIBMGA010024	2.37	Serine protease activity
	27	H9JG67	BGIBMGA008514	2.02	Serine protease activity
	28	H9JY09	BGIBMGA014427	2.44	Serine protease activity
	29	H9J231	BGIBMGA003568	2.72	Serine protease activity
	30	H9JJ26	BGIBMGA009527	2.64	Serine protease activity
	31	H9JIY5	BGIBMGA009486	2.33	Carboxypeptidase activity
	32	H9JES0	BGIBMGA008017	2.37	AMP deaminase activity
	33	H9JEW2	BGIBMGA008059	4.47	Aminopeptidase activity
	34	H9JDE9	BGIBMGA007546	3.86	Carboxylesterase activity
	35	H9JE19	BGIBMGA007766	2.35	Phosphoric diester hydrolase activity
	36	B5TZ28	BGIBMGA007915	3.13	Metallopeptidase activity
	37	H9IWJ8	BGIBMGA001630	2.01	Cholinesterase activity
	38	H9JSJ8	BGIBMGA012509	3.51	Glucosinolate sulphatase activity
	39	H9J064	BGIBMGA002899	2.18	Carboxylesterase activity
	40	H9J067	BGIBMGA002902	3.26	Cholinesterase activity
	41	B2ZDZ0	BGIBMGA009544	2.04	Carboxylesterase activity
	42	C0SQ80	BGIBMGA008354	2.00	Odorant binding
	43	A1YQ87	BGIBMGA005493	3.08	Phosphopyruvate hydratase activity
	44	H9JP12	BGIBMGA011266	3.08	Insect hexamerins
	45	H9J128	BGIBMGA003215	4.77	RNA binding
	46	H9JLC5	BGIBMGA010326	3.09	Mitochondrial carriers
	47	H9IVY4	BGIBMGA001415	3.89	Ubiquitin-mediated protein binding
	48	H9IT95	BGIBMGA000475	5.80	Cation binding
	49	Q69FX2	BGIBMGA008221	2.34	Innate immunity and lipid metabolism
	50	S5M110	BGIBMGA005577	2.25	Carbohydrate derivative binding
	51	C1K001	BGIBMGA004287	3.16	N/A
	52	H9JYG4	BGIBMGA009648	5.55	DNA binding
	53	H9IYN2	BGIBMGA002366	2.16	N/A
	54	Q2F645	BGIBMGA014211	2.03	Transketolase activity
	55	H9J1D5	BGIBMGA003322	4.40	Juvenile hormone binding
	56	H9J5L9	BGIBMGA004809	3.95	Lyase activity
	57	H9J1D8	BGIBMGA003325	4.85	N/A
	58	H9JT75	BGIBMGA012737	2.58	Peroxidase activity
	59	H9JXD9	BGIBMGA014204	2.02	N/A
	60	H9JPZ3	BGIBMGA011598	3.59	N/A
	61	H9J3M9	BGIBMGA004116	2.02	Transferase activity
	62	H9J1D9	BGIBMGA003326	8.57	N/A
	63	H9JXN1	BGIBMGA014298	12.23	N/A
DJ48/24 h					
	1	H9IW76	BGIBMGA001507	2.87	Lipase
	2	C7EPE2	BGIBMGA000158	4.05	Glucose-methanol-choline (GMC) oxidoreductase activity
	3	H9JFH1	BGIBMGA008268	2.12	Aldehyde oxidase activity
	4	H9ISL2	BGIBMGA000239	2.30	Peroxidase activity
	5	H9ISL1	BGIBMGA000238	4.67	Peroxidase activity
	6	H9IX93	BGIBMGA001876	2.52	Alpha-amylase activity
	7	B2DD57	BGIBMGA005696	2.01	Glycosyl hydrolase activity
	8	A0A077JI83	BGIBMGA006066	3.67	O-Glycosyl hydrolase activity
	9	A1YQ87	BGIBMGA005493	3.06	Phosphopyruvate hydratase activity
	10	P81902	—	2.03	Trypsin inhibitor
	11	C0J8H1	—	2.02	N/A
	12	H9JKL1	BGIBMGA010062	3.09	Serine-type endopeptidase activity
	13	H9J5K7	BGIBMGA004797	2.14	Metallocarboxypeptidase activity
	14	H9JG67	BGIBMGA008514	2.28	Serine protease activity
	15	H9J5P0	BGIBMGA004830	2.12	Metallocarboxypeptidase activity
	16	H9JIY5	BGIBMGA009486	2.41	Carboxypeptidase activity
	17	H9JG68	BGIBMGA008515	4.20	Serine protease activity
	18	H9JHZ0	BGIBMGA009138	2.00	Aminopeptidase activity
	19	A7LIK7	BGIBMGA004403	2.14	30K protein
	20	H9J128	BGIBMGA003215	2.72	RNA binding
	21	H9JLC5	BGIBMGA010326	3.06	Mitochondrial carriers
	22	H9IVY4	BGIBMGA001415	2.24	Ubiquitin-mediated protein binding
	23	H9JUE4	BGIBMGA013157	3.54	Ribosomal protein
	24	Q8N0P2	BGIBMGA002381	3.43	70 kilodalton heat shock protein
	25	H9J5L9	BGIBMGA004809	2.80	Lyase activity
	26	H9J1D9	BGIBMGA003326	3.69	N/A
	27	H9JXN1	BGIBMGA014298	6.68	N/A

“—” indicates that no SilkDB accession BGI number was found. N/A: not applicable.

**Table 2 ijms-20-06113-t002:** Top 10 of the KEGG pathway enrichment analysis of DEPs in DJ24h compared with DJ0h.

Number	Map_Name	Map_ID	Protein_ID	Definition	Fold Change
1	Galactose metabolism	map00480	F8V3L0	gamma-glutamyltranspeptidase [24]	0.47
2	Lysosome	map04142	H9JSN7	sphingomyelin phosphodiesterase [25]	3.21
3	Sphingolipid metabolism	map00600	H9JSN7	sphingomyelin phosphodiesterase [26]	3.21
4	Arachidonic acid metabolism	map00590	F8V3L0	gamma-glutamyltranspeptidase [24]	0.47
5	Taurine and hypotaurine metabolism	map00430	F8V3L0	gamma-glutamyltranspeptidase [24]	0.47
6	Neuroactive ligand-receptor interaction	map04080	H9JL75	trypsin [27]	3.15
7	Folate biosynthesis	map00790	B2ZZX0	alkaline phosphatase [28]	4.18
8	Thiamine metabolism	map00730	B2ZZX0	alkaline phosphatase [28]	4.18
9	Ubiquitin mediated proteolysis	map04120	H9JK60	ubiquitin-activating enzyme E1 [29]	2.05
10	Caffeine metabolism	map00232	H9JFX4	xanthine dehydrogenase/oxidase [30]	0.28

**Table 3 ijms-20-06113-t003:** Top 10 of the KEGG pathway enrichment analysis of DEPs in DJ48h compared with DJ24h.

Number	Map_Name	Map_ID	Protein_ID	Definition	Fold Change
1	Folate biosynthesis	map00790	H9IVS0	aldehyde reductase [31]	3.11
			B2ZZX0	alkaline phosphatase [28]	4.18
			H9JTG9	aldehyde reductase [32]	0.50
			H9IVT6	aldehyde reductase	0.06
2	Lysosome	map04142	Q69FX2	niemann-Pick C2 protein [33,34]	2.34
			A4PHN6	hexosaminidase [35]	0.50
			H9IWR3	lysosomal alpha-mannosidase [36]	0.47
3	Galactose metabolism	map00052	H9IVS0	aldehyde reductase [37]	3.11
			H9JTG9	aldehyde reductase [32]	0.50
			H9IVT6	aldehyde reductase	0.06
4	Fructose and mannose metabolism	map00051	H9IVS0	aldehyde reductase [37]	3.11
			H9JTG9	aldehyde reductase [32]	0.50
			H9IVT6	aldehyde reductase	0.06
5	Pentose and glucuronate interconversions	map00040	H9IVS0	aldehyde reductase [37]	3.11
			H9JTG9	aldehyde reductase [32]	0.50
			H9IVT6	aldehyde reductase	0.06
6	Glycerolipid metabolism	map00561	H9IVS0	aldehyde reductase [37]	3.11
			H9JTG9	aldehyde reductase [32]	0.50
			H9IVT6	aldehyde reductase	0.06
7	Purine metabolism	map00230	H9JFH1	xanthine dehydrogenase/oxidase [30]	3.06
			H9JDV4	nucleoside-diphosphate kinase [38]	0.26
			H9JFX4	xanthine dehydrogenase/oxidase [30]	0.38
8	Peroxisome	map04146	H9JFH1	xanthine dehydrogenase/oxidase [30]	3.06
			Q08J22	superoxide dismutase, Cu-Zn family [39,40]	0.33
			H9JFX4	xanthine dehydrogenase/oxidase [30]	0.38
9	Starch and sucrose metabolism	map00500	H9IX93	alpha-amylase [41]	2.78
			H9J822	alpha-trehalase [42]	0.42
10	Glutathione metabolism	map00480	H9JES0	aminopeptidase N [43]	2.37
			H9JEW2	aminopeptidase N [43]	4.47

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
