# Peer review of "iTRAQ-Based Quantitative Proteomic Analysis of Digestive Juice across the First 48 Hours of the Fifth Instar in Silkworm Larvae"

_ijms, 2019, doi:10.3390/ijms20246113_

Round 1
Reviewer 1 Report
Pingzhen Xu, etal., did proteomic analysis of digestive juice from frist 48 h of 5th instar silkworm. They identified 227 proteins totally. 30 genes encoding the identified proteins were of tissue-specific expression in the midgut. They also did bioimformatic analysis of identified protein. They verified the change of mRNA levels of identified protein genes in midgut. Their study will provide improtant imformation for further studies on the silwkorm developmemt and proteomics. However, before publication, there are several questions needed to be addressed.
Major
1. Why did not you include samples from the time point, 72h of the 5th instar? I think it is important, because on day 3 (48h to 72h), peak amount of protiens may be secreted or prodoced digestive juice from midgut, as stated that Day three of the fifth instar of the silkworm is the boundary for whole larval development stage (line 201).
2. Can you make a figure to show digestive juice and intestines from 0h, 24h, 48h of the 5th instar silkworms, respectively?
3. Fig. 4, the micro array data is raw. I think it will be much better to calculate the mean value of each gene in each titues. Then, re-make this figure to show more concise and clear data.
Minor
The title is too long and contains redundant words, like Temporal vs First Three Days. Suggestion is "iTRAQ-based quantitative proteomic analysis of digestive juice across first 48 hours of the 5th instar in silkworm larvae"
Line 26, what is "resisting inhibitions"?
Line 137, immunity of B. mori larvae are gradually enhanced. In section 2.1 and 2.2, you did not mention any proteins are invloved in silkworm immunity. Can you anlyze or categorize any proteins identifed to immunity? If you can, you need to show this result, that will be very interesting.
Line 183, protein–protein interactions (Figure 4) to protein–protein interactions (Figure 3)
Fig. 5, why did you choose BmRPL3 as internal control gene rather than actin3 or GAPDH?
Author Response
Response to Reviewer 1 Comments
Comments and Suggestions for Authors:
Pingzhen Xu, etal., did proteomic analysis of digestive juice from frist 48 h of 5th instar silkworm. They identified 227 proteins totally. 30 genes encoding the identified proteins were of tissue-specific expression in the midgut. They also did bioimformatic analysis of identified protein. They verified the change of mRNA levels of identified protein genes in midgut. Their study will provide improtant imformation for further studies on the silwkorm developmemt and proteomics. However, before publication, there are several questions needed to be addressed.
Response: Thanks for the reviewer’s good evaluation and kind suggestion. The advices from you have provided with great helps to us in the current work. Due to your suggestion, we carefully revised our manuscript, If the responses are not accurate and/or in place, please give criticism and more advices. Thank you very much for your careful reviews in our work.
Major points
Point 1: Why did not you include samples from the time point, 72h of the 5th instar? I think it is important, because on day 3 (48h to 72h), peak amount of protiens may be secreted or prodoced digestive juice from midgut, as stated that Day three of the fifth instar of the silkworm is the boundary for whole larval development stage (line 201).
Response 1: Thanks for the reviewer’s good evaluation and kind suggestion. Day three (48h to 72h) of the fifth instar of the silkworm is an important time-point. The following factors were taken into account when we designed the study protocol. (ⅰ): silkworms have entered the stage of rapid growth from feeding 48 h of fifth instar. (ⅱ): the amount of leaf ingested and digested quickly increases from 0h to 72h after the fifth instar ecdysis. (ⅲ): the amount of leaf ingested and digested is relatively stable from 72h to 120h after the fifth instar ecdysis, this is largely due to the fact that the midgut volume has peaked. (ⅳ): the percentage of leaf digested quickly increases from 0h to 48h after the fifth instar ecdysis, but after 48h it gradually decreases. So we collected the digestive juice from 0h to 48h after the fifth instar ecdysis. It is also important of the time point on day 3 (48h to 72h). In the design of the study protocol, whether to include this time point, we were very tangled. The careful and precise reviews, and the strategic and constructive advices from you have provided with great helps to us in the current work and will improve our level of scientific research in the future work. In our future work on comparative proteomics of digestive juice in silkworm fed by optimum ripe mulberry leaves and aged mulberry leaves (distinguished in sections of the tree), and digestive juice in silkworm within large amount of leaf ingested and high percentage of leaf ingested, we have obtained the inspirations and ideas from your strategic and constructive suggestions. Thank you very much.
Point 2: Can you make a figure to show digestive juice and intestines from 0h, 24h, 48h of the 5th instar silkworms, respectively?
Response 2: Thanks for the reviewer’s kind suggestion. Digestive juice was collected by electric shock (35 V). The gut juice was centrifuged at 1500g to remove undigested mulberry leaves. The midgut was washed in 0.85% NaCl three times, and was collected to a new tube.
Point 3: Fig. 4, the micro array data is raw. I think it will be much better to calculate the mean value of each gene in each tissues. Then, re-make this figure to show more concise and clear data.
Response 3: Thanks for the reviewer’s good advice and kind suggestion. According to your suggestion and the characteristics of the expression of midgut specific genes, the genes significantly high expressed in midgut comparing with other tissues, and the other genes expressed and/or unexpressed in all tissues, we re-maked this figure. The expression of midgut specific genes is shown in Figure. 4A, the genes significantly high expressed in midgut comparing with other tissues is shown in Figure. 4B, and and the other genes expressed and/or unexpressed in all tissues is shown in Figure. S3. The precise reviews and the constructive advices from you have provided with great helps to us in the current work. Thank you very much.
Minor points
Point 4: The title is too long and contains redundant words, like Temporal vs First Three Days. Suggestion is "iTRAQ-based quantitative proteomic analysis of digestive juice across first 48 hours of the 5th instar in silkworm larvae"
Response 4: Thanks for the reviewer’s good evaluation. Due to your evaluation, we have changed the title of our manuscript to "iTRAQ-based quantitative proteomic analysis of digestive juice across first 48 hours of the 5th instar in silkworm larvae". The careful and precise reviews, and the constructive advices from you have provided with great helps to us in the current work. Thank you very much.
Point 5: Line 26, what is "resisting inhibitions"?
Response 5: Thanks for the reviewer’s good evaluation and kind suggestion. In silkworm digestive juice, different counter adaptive strategies including the overproduction of active proteases to overrule the inhibitory effects of mulberry leaves protease inhibitors (PIs). We have used "overruling the inhibitory effects of mulberry leaves protease inhibitors (PIs)" to replace "resisting inhibitions" in the revised version. It can be found in Line 23. Thank you very much for your careful reviews in our work.
Point 6: Line 137, immunity of B. mori larvae are gradually enhanced. In section 2.1 and 2.2, you did not mention any proteins are invloved in silkworm immunity. Can you anlyze or categorize any proteins identifed to immunity? If you can, you need to show this result, that will be very interesting.
Response 6: Thanks for the reviewer’s kind suggestion. Five serine proteases were identified and increased in digestive juice. They are associated with immune response during infection. The transcriptional levels of the five genes encoding serine proteases were up-regulated via RT-qPCR analysis during Escherichia coli and Bacillus bombyseptieus infection, respectively. The results of RT-qPCR analysis have been added in the revised version with the attached "Figure S1".
Point 7: Line 183, protein–protein interactions (Figure 4) to protein–protein interactions (Figure 3)。
Response 7: Thanks for the reviewer’s careful reviews in our work. We have corrected this mistake.
Point 8: Fig. 5, why did you choose BmRPL3 as internal control gene rather than actin3 or GAPDH?
Response 8: Thanks for the reviewer’s kind suggestion. BmRPL3, actin3 and GAPDH can be selected reference genes in RT-qPCR analysis in silkworm. Stability of BmRPL3 is impartial that was determined in normal tissues using geNorm and NormFinder. The careful and precise reviews, and the constructive advices from you have provided with great helps to us in the current work. Thank you very much.

Reviewer 2 Report
This is a manuscript dealing with interesting subjects, results and conclusions. However, before acceptance (if this is the case) it should be revised in depth regarding, at least, the english style (as in the construction of sentences, i.e. at lines 54-55, 62-63 or others), the excessive length of tables (mainly table 1), of references (148) and of text, and the quality of figures (as in Figs. 1A-B, 2A-B, 4, and 5, mending the repeated use of very small characters, which makes difficult its reading and understanding).
Besides, the manuscript should be improved in other more particular aspects:
-The way to deal and explain functional assignments and locations of identified proteins is frequently too heterogeneous, with an excessive use of generic terms (i.e. cell, cell part, binding, involved in pathways ...), mixing them with more specific and informative terms (extracellular region, peroxisome, phosphatidylethanolamide binding, folate biosynthesis ...), which is a more informative and desirable way. A functional assignment like "proteolysis" surely will be useless in many cases, but another one like "serine protease activity" more informative and useful. Also, the use of internal or nearby redundancies should be avoided, like "proteases with hydrolase activity"... To the knowledge of this reviewer a protease has an intrinsic hydrolase activity (unless inactive) and this use is incorrect. The same happens in the list of functions (i.e. in Table 1) in which terms as hydrolase activity, proteolysis, serine protease activity (or others) .... etc, of distinct levels and definition, are mixed.
-Authors should make clear which publications included in the paper come from their own laboratory/group and which ones are external, something that is not clear enough now. More definition is also required regarding the origin of certain results discussed in the paper; the most demanding is related with the widely used and discussed tissue expressed profiles of genes (arrays) of digestive juice proteins which results are not clearly indicated whether have been derived in the present publication or in previous one(s). This is very important.
-The concept of alkaline proteins in the digestive juice (70 identified as such and 20 more as near alkaline), mentioned in different parts of the paper, which also were referred as more stable in alkaline conditions (lines 249-250), could be misleading. Is this an experimental observation or a thought of the authors? Could it be supported by references? Given that the silkworm larvae juice is alkaline, the stability of its proteins in such condition not necessarily would be more stable but certainly less soluble, given its less charged state (nearby isoelectric point). By the way, authors should indicate more precisely in the paper which is the definite pH of such juice, or the known range of pHs at the different parts of the gut. The term "alkaline" or "near alkaline" is excessively undefined.
-The manuscript should also be revised for typing mistakes or lack of definitions, which occur in different parts of the paper: i.e. at line 66 it should read, "...the digestive enzymes trypsins, chymotrypsins ...", at line 97 it should read "... 1126 unique tryptic peptides ...", at line 202 should it read "... to synthesize mass silk ..." or "to synthesize massively silk ...", at line 264 it should read "... carbohydrates and lipids ...", at line 363 it should read "... metabolites and protease inhibitors ..." (proteinase only refers to endopeptidases!) , etc etc. Besides, at Conclusions, at line 449, it is said that serine protease inhibitor activity is involved in the digestion of mulberry leaves, something that occurs unlikely and the sentence should be reformulated.
Author Response
Response to Reviewer 2 Comments
Comments and Suggestions for Authors:
This is a manuscript dealing with interesting subjects, results and conclusions. However, before acceptance (if this is the case) it should be revised in depth regarding, at least, the english style (as in the construction of sentences, i.e. at lines 54-55, 62-63 or others), the excessive length of tables (mainly table 1), of references (148) and of text, and the quality of figures (as in Figs. 1A-B, 2A-B, 4, and 5, mending the repeated use of very small characters, which makes difficult its reading and understanding).
Response: Thanks for the reviewer’s good evaluation and kind suggestion. The advices from you have provided with great helps to us in the current work. Due to your suggestion, we carefully revised our manuscript, If the responses are not accurate and/or in place, please give criticism and more advices. Thank you very much for your careful reviews in our work.
Major points
Point 1: Besides, the manuscript should be improved in other more particular aspects:
-The way to deal and explain functional assignments and locations of identified proteins is frequently too heterogeneous, with an excessive use of generic terms (i.e. cell, cell part, binding, involved in pathways ...), mixing them with more specific and informative terms (extracellular region, peroxisome, phosphatidylethanolamide binding, folate biosynthesis ...), which is a more informative and desirable way. A functional assignment like "proteolysis" surely will be useless in many cases, but another one like "serine protease activity" more informative and useful. Also, the use of internal or nearby redundancies should be avoided, like "proteases with hydrolase activity"... To the knowledge of this reviewer a protease has an intrinsic hydrolase activity (unless inactive) and this use is incorrect. The same happens in the list of functions (i.e. in Table 1) in which terms as hydrolase activity, proteolysis, serine protease activity (or others) .... etc, of distinct levels and definition, are mixed.
Response 1: Thanks for the reviewer’s good evaluation and kind suggestion. Due to your suggestion, (ⅰ): we carefully revised our manuscript including removing the redundant words, sentence clearness and smoothness, punctuation and spelling, etc; (ⅱ): we simplified the tables. The section of proteins down-regulated in previous Table 1 was removed as shown in Table S2, the order of other “Supplementary Table” was adjusted accordingly. The top 10 of the KEGG pathways were selected and made into new tables 2 and 3; (ⅲ): the number of references has shrunk; (ⅳ): the quality of figures was improved in the revised version. The revisions are highlighted using the "Red Font" and "Track Changes" function in Microsoft Word. Thank you very much for your careful reviews in our work.
Point 2: -Authors should make clear which publications included in the paper come from their own laboratory/group and which ones are external, something that is not clear enough now. More definition is also required regarding the origin of certain results discussed in the paper; the most demanding is related with the widely used and discussed tissue expressed profiles of genes (arrays) of digestive juice proteins which results are not clearly indicated whether have been derived in the present publication or in previous one(s). This is very important.
Response 2: Thanks for the reviewer’s good evaluation and kind suggestion. According to your advices, we carefully revised our manuscript including a clear presentation of our results and discussion, concise and accurate wording, and sentence smoothness, etc. We re-maked the Figure 4 to show more concise and clear data. The microarray data is raw. We calculated the mean value of each gene in each tissue. The characteristics of the spatial expression profile can be divided into three parts: (ⅰ) the expression of midgut specific genes, (ⅱ) the genes significantly high expressed in midgut comparing with other tissues, (ⅲ) and the other genes expressed and/or unexpressed in all tissues. The expression of midgut specific genes is shown in Figure. 4A, the genes significantly high expressed in midgut comparing with other tissues is shown in Figure. 4B, and and the other genes expressed and/or unexpressed in all tissues is shown in Figure. S3. The corresponding revisions are highlighted using the "Red Font" and "Track Changes" function in Microsoft Word. The precise reviews and the constructive advices from you have provided with great helps to us in the current work. Thank you very much.
Point 3: -The concept of alkaline proteins in the digestive juice (70 identified as such and 20 more as near alkaline), mentioned in different parts of the paper, which also were referred as more stable in alkaline conditions (lines 249-250), could be misleading. Is this an experimental observation or a thought of the authors? Could it be supported by references? Given that the silkworm larvae juice is alkaline, the stability of its proteins in such condition not necessarily would be more stable but certainly less soluble, given its less charged state (nearby isoelectric point). By the way, authors should indicate more precisely in the paper which is the definite pH of such juice, or the known range of pHs at the different parts of the gut. The term "alkaline" or "near alkaline" is excessively undefined.
Response 3: Thanks for the reviewer’s good evaluation and kind advice. A total of 227 proteins were successfully identified. Among them, 70 proteins were alkaline and 20 proteins were nearly alkaline at the isoelectric point. We mentioned of "which were stable in the alkaline environment of digestive juice" that is inaccurate. We are sorry. Due to your evaluation, we have corrected this error and the contents of the range of pHs at the different parts of the gut are added in "Introduction" section in the revised version. The corresponding revisions are highlighted using the "Red Font" and "Track Changes" function in Microsoft Word. Thank you very much for pointing out mistakes so that they can be corrected.
Point 4: -The manuscript should also be revised for typing mistakes or lack of definitions, which occur in different parts of the paper: i.e. at line 66 it should read, "...the digestive enzymes trypsins, chymotrypsins ...", at line 97 it should read "... 1126 unique tryptic peptides ...", at line 202 should it read "... to synthesize mass silk ..." or "to synthesize massively silk ...", at line 264 it should read "... carbohydrates and lipids ...", at line 363 it should read "... metabolites and protease inhibitors ..." (proteinase only refers to endopeptidases!) , etc etc. Besides, at Conclusions, at line 449, it is said that serine protease inhibitor activity is involved in the digestion of mulberry leaves, something that occurs unlikely and the sentence should be reformulated.
Response 4: Thanks for the reviewer’s good evaluation and kind suggestion. We carefully revised our manuscript including sentence smoothness, punctuation and spelling, etc. The corresponding revisions are highlighted using the "Red Font" and "Track Changes" function in Microsoft Word. The advices from you have provided with great helps to us in the current work. Thank you very much.

Reviewer 3 Report
In this study Xu et al carried out the proteomic analysis for the silkworm digestive juice to clarify the developmental dynamics of the digestion of the mulberry. To date there have been a number of studies that carried out the genomics and/or transcriptomics in the silkworm but the proteomic study is still very limited. The study performed by the authors are of significance in this point. The authors challenged to elucidate the mechanism of the food digestion process, one of the most important topics in the silkworm biology, and found a number of proteins that could function in this process. Especially, the analysis in several different developmental stages should be a significant milestone to understand the complex dynamics of this biological process. The experiment has been done adequately and the manuscript is written with clear English. I consider after revisions described below the manuscript can be acceptable.
<Major points>
1.The discussion is descriptive, just stating the previous studies relevant for the identified proteins. This is not “discussion”. Also, this section is too long. The authors should consider again completely for this part. For example, how about discussing the biological meaning of the proteins expressed differently between stages?
2.Line 114. Is there any reason that the authors set the threshold of the fold change as “2.0” for the up-regulation and “0.5” for the down-regulation?
3.Figure 3. This figure is shown with Swiss-Prot accession number but these IDs are not described in the manuscript (just the gene number (BGIBMGA) is shown, except for the H9IVY4). This is confusing.
4.Figure 4. This figure is too small to see. I propose that the whole heat map is shown in the supplemental figure and just important part is shown here. For example, the expression of midgut specific genes is shown in Fig. 4A and those expressed in all tissues are shown in Fig. 4B.
<Minor points>
1.Line 23. of tissue-specific expression in the midgut -> expressed specifically in the midgut
2.Line 51. “The first is exoenzymes that alkaline conditions are mostly adopted” <- What does this mean?
3.Line 62. “specifically a hydrolyzing substrate of beta-naphthylacetate” <- What does this mean?
4.Line 183. Figure 4 -> Figure 3.
5. Line 207. The genes had expression signals -> Genes of the identified digestive juice proteins had expression signals
Author Response
Response to Reviewer 3 Comments
Comments and Suggestions for Authors:
In this study Xu et al carried out the proteomic analysis for the silkworm digestive juice to clarify the developmental dynamics of the digestion of the mulberry. To date there have been a number of studies that carried out the genomics and/or transcriptomics in the silkworm but the proteomic study is still very limited. The study performed by the authors are of significance in this point. The authors challenged to elucidate the mechanism of the food digestion process, one of the most important topics in the silkworm biology, and found a number of proteins that could function in this process. Especially, the analysis in several different developmental stages should be a significant milestone to understand the complex dynamics of this biological process. The experiment has been done adequately and the manuscript is written with clear English. I consider after revisions described below the manuscript can be acceptable.
Response: Thanks for the reviewer’s good evaluation and kind suggestion. The advices from you have provided with great helps to us in the current work. Due to your suggestion, we carefully revised our manuscript, If the responses are not accurate and/or in place, please give criticism and more advices. Thank you very much for your careful reviews in our work.
Major points
Point 1: The discussion is descriptive, just stating the previous studies relevant for the identified proteins. This is not “discussion”. Also, this section is too long. The authors should consider again completely for this part. For example, how about discussing the biological meaning of the proteins expressed differently between stages?
Response 1: Thanks for the reviewer’s good evaluation and kind suggestion. the section of “Discussion” has also been supplemented, adjusted and improved. The revisions are highlighted using the "Red Font" and "Track Changes" function in Microsoft Word. Thank you very much for your careful reviews in our work.
Point 2: Line 114. Is there any reason that the authors set the threshold of the fold change as “2.0” for the up-regulation and “0.5” for the down-regulation?
Response 2: Thanks for the reviewer’s kind suggestion. In the study of comparative proteomics, the threshold of the fold changes of ≥2.5, ≥2.0, ≥1.5 or ≤0.4, ≤0.5, ≤0.667 were commonly used to identify differently expressed protein. So we set the threshold of the fold changes as “2.0” for the up-regulation protein identified and “0.5” for the down-regulation protein identified.
Point 3: Figure 3. This figure is shown with Swiss-Prot accession number but these IDs are not described in the manuscript (just the gene number (BGIBMGA) is shown, except for the H9IVY4). This is confusing.
Response 3: Thanks for the reviewer’s good evaluation and kind suggestion. We have added the Swiss-Prot accession number of each protein in the revised version.
Point 4: Figure 4. This figure is too small to see. I propose that the whole heat map is shown in the supplemental figure and just important part is shown here. For example, the expression of midgut specific genes is shown in Fig. 4A and those expressed in all tissues are shown in Fig. 4B.
Response 4: Thanks for the reviewer’s good advice and kind suggestion. According to your suggestion, we re-maked this figure to show more concise and clear data. The microarray data is raw. We calculated the mean value of each gene in each tissue. The expression of midgut specific genes is shown in Figure. 4A, the genes significantly high expressed in midgut comparing with other tissues is shown in Figure. 4B, and and the other genes expressed and/or unexpressed in all tissues is shown in Figure. S3. The precise reviews and the constructive advices from you have provided with great helps to us in the current work. Thank you very much.
Minor points
Point 5: Line 23. of tissue-specific expression in the midgut -> expressed specifically in the midgut.
Response 5: Thanks for the reviewer’s good evaluation. We have used "expressed specifically in the midgut" to replace "of tissue-specific expression in the midgut" in the revised version. It can be found in Line 24. Thank you very much for your careful reviews in our work.
Point 6: Line 51. “The first is exoenzymes that alkaline conditions are mostly adopted” <- What does this mean?
Response 6: Thanks for the reviewer’s good evaluation and kind suggestion. We have used "The first is exoenzymes that are synthesized by epithelial cells and involved in the degradation of macromolecular nutrients, such as trypsin, lipase, amylase, and nuclease, the degradation processes under alkaline conditions are mostly adopted as suitable enzymatic conditions" to replace "The first is exoenzymes that alkaline conditions are mostly adopted, such as trypsin, lipase, amylase, and nuclease, which are synthesized by epithelial cells and involved in the degradation of macromolecular nutrients" in the revised version. It can be found in Line 55-58, page 2.
Point 7: Line 62. “specifically a hydrolyzing substrate of beta-naphthylacetate” <- What does this mean?
Response 7: Thanks for the reviewer’s good evaluation. We are sorry. The description makes difficult to read and understand. We have used "The esterases, specific β-esterase bands (Est-1, 2 and 3), have been documented as being present in digestive juice" to replace "The esterases, specifically a hydrolyzing substrate of β-naphthylacetate (Est-1, 2 and 3), have been documented as being present in digestive juice" in the revised version. It can be found in Line 66-67, page 2.
Point 8: Line 183. Figure 4 -> Figure 3.
Response 8: Thanks for the reviewer’s careful reviews in our work. We have corrected this mistake.
Point 9: Line 207. The genes had expression signals -> Genes of the identified digestive juice proteins had expression signals.
Response 9: Thanks for the reviewer’s good evaluation. According to your advices, we have used "Genes of the identified digestive juice proteins had expression signals in at least one tissue" to replace "The genes had expression signals in at least one tissue" in the revised version. The careful and precise reviews, and the constructive advices from you have provided with great helps to us in the current work. Thank you very much.

Round 2
Reviewer 1 Report
Basically, the revised version is improved accordingly.
For point 2 to show a digestive juice and intestines of silkworm samples, I guess you can not get fresh silkworm samples in this season. It is fine.
For point 8, Stability of BmRPL3 is impartial that was determined in normal tissues using geNorm and NormFinder. Do you have any references to support this statement, since you used action 3 in you previous report (line 442, citation [96]).
Author Response
Response to Reviewer 1 Comments Comments and Suggestions for Authors: Basically, the revised version is improved accordingly. Response: Thank you very much for your recognition and support. Our deepest tribute and thanks go to you. Point 1: For point 2 to show a digestive juice and intestines of silkworm samples, I guess you can not get fresh silkworm samples in this season. It is fine. Response 1: Thank you very much for your understanding and support. You are familiar with our sericulture and silkworm breeding. Indeed, there were no mulberry leaves in this season in our locality, and silkworms could not be raised. You are a considerate and tolerant person, providing with careful and precise reviews, and the strategic and constructive advices for our work. And for that, we would like to our heartfelt gratefulness. Point 2: For point 8, Stability of BmRPL3 is impartial that was determined in normal tissues using geNorm and NormFinder. Do you have any references to support this statement, since you used action 3 in your previous report (line 442, citation [96]). Response 2: Thanks for the reviewer’s good advice and kind suggestion. We are sorry, we don't have references to support the statement. The first author, while a phd student (studying in state key laboratory of silkworm genome biology, Southwest University), understood that BmRPL3 could be used as a housekeeping gene for the development of silkworm head, epidermis, midgut and trachea through their own analysis (not published). The team at Southwest University is also continuing to use BmRPL3, as they published papers (Repression of tyrosine hydroxylase is responsible for the sex-linked chocolate mutation of the silkworm, Bombyx mori, PNAS, 2010, doi: 10.1073/pnas.1001725107. Epub 2010 Jul 6; BmBlimp-1 gene encoding a C2H2 zinc finger protein is required for wing development in the silkworm Bombyx mori, Int J Biol Sci, 2019, doi: 10.7150/ijbs.34743. eCollection 2019). Therefore, we also used BmRPL3 as an internal control to normalize for equal sample loading. The precise reviews from you have provided with great helps to us in the current work. Thank you very much.Reviewer 3 Report
1.Line 55. The first is exoenzymes … such as trypsin, lipase, amylase, and nuclease, the degradation processes under alkaline conditions are mostly adopted as suitable enzymatic conditions.
->The first is exoenzymes … such as trypsin, lipase, amylase, and nuclease. These enzymes can work efficiently under the alkaline conditions.
2.Again, I propose to revise the discussion. It can be compacted more and more. The authors should focus on “digestion enzymes” and “proteins involved in immunity” and in this aspect, Line 267-322 can be summarized as one paragraph and the last paragraph (Line 367-388) can be removed.
Author Response
Response to Reviewer 3 Comments Point 1: 1.Line 55. The first is exoenzymes … such as trypsin, lipase, amylase, and nuclease, the degradation processes under alkaline conditions are mostly adopted as suitable enzymatic conditions. ->The first is exoenzymes … such as trypsin, lipase, amylase, and nuclease. These enzymes can work efficiently under the alkaline conditions. Response 1: Thanks for the reviewer’s good evaluation and kind suggestion. The revisions are highlighted using the "Red Font" and "Track Changes" function in Microsoft Word. It can be found in Line 56-57, page 2. Thank you very much for your careful reviews in our work. Point 2: 2. Again, I propose to revise the discussion. It can be compacted more and more. The authors should focus on “digestion enzymes” and “proteins involved in immunity” and in this aspect, Line 267-322 can be summarized as one paragraph and the last paragraph (Line 367-388) can be removed. Response 2: Thanks for the reviewer’s good advice and kind suggestion. According to your suggestion, in the revised version, (ⅰ): the part of "Discussion" has been compacted focusing on the themes of "digestion enzymes" and "proteins involved in immunity"; (ⅱ): the part of Line 367-388 has been removed; (ⅲ): the part of Line 367-388 has been summarized as one paragraph, the word count has been reduced from 736 to 521. The constructive advices from you have provided with great helps to us in the current work. Thank you very much.